# 2D and 3D Triangulation Are Suitable In Situ Measurement Tools for High-Power Large Spot Laser Penetration Processes to Visualize Depressions and Protrusions before Perforating

**DOI:** 10.3390/ma15113743

**Published:** 2022-05-24

**Authors:** Stefan Reich, Alexander Göbel, Marcel Goesmann, Dominic Heunoske, Sebastian Schäffer, Martin Lueck, Matthias Wickert, Jens Osterholz

**Affiliations:** Fraunhofer Institute for High-Speed Dynamics, Ernst–Mach–Institut, EMI, Ernst-Zermelo Straße 4, 79104 Freiburg, Germany; alexanderxgoebel@web.de (A.G.); marcel.goesmann@emi.fraunhofer.de (M.G.); dominic.heunoske@emi.fraunhofer.de (D.H.); sebastian.schaeffer@emi.fraunhofer.de (S.S.); martin.lueck@emi.fraunhofer.de (M.L.); matthias.wickert@emi.fraunhofer.de (M.W.); jens.osterholz@emi.fraunhofer.de (J.O.)

**Keywords:** high-power laser, laser penetration, 2D triangulation

## Abstract

During laser penetration, the irradiated samples form a melt pool before perforation. Knowledge of the dynamics of this melt pool is of interest for the correct physical description of the process and leads to improved simulations. However, a direct investigation, especially at the location of high-power laser interaction with large spot diameters in the centimeter range is missing until now. Here, the applicability of 2D triangulation for surface topology observations is demonstrated. With the designed bidirectional 2D triangulation setup, the material cross-section is measured by profile detection at the front and back side. This allows a comprehensive description of the penetration process to be established, which is important for a detailed explanation of the process. Specific steps such as surface melting, indentations, protrusions during melt pool development and their dynamics, and the perforation are visualized, which were unknown until now. Furthermore, a scanning 3D triangulation setup is developed to obtain more information about the entire melt pool at the front side, and not just a single intersection line. The measurements exhibit a mirror-symmetric melt pool and the possibility to extrapolate from the central profile to the outer regions in most cases.

## 1. Introduction

Lasers are currently widely used in material processing, such as cutting, welding, or surface modification [1,2]. The used laser spot sizes are in the range of sub-millimeters up to some millimeters. As there is a large industrial usage, the underlying processes of heating, melting, and also melt pool dynamics are well known [3,4,5]. However, not much is known for large laser spots. There are few experimental investigations of large laser spot penetration processes [6,7,8], as well as some simulations [9].

An in situ monitoring of the melt pool during laser processing is challenging due to the very harsh conditions [5]. Only indirect or optical measurements are possible. From optical imaging, certain aspects of the melt pool generation and also dynamics might be extractable [10]. However, obtaining 3D information from a 2D image is always limited, especially if the viewing angle is restricted. In addition, a high dynamic range of intensities can easily occur during laser processing, making imaging even more challenging. An imaging technique not disturbed by any light emissions from laser processing is X-ray imaging [11,12]. From the X-ray absorption contrast, the material thickness can be calculated. Depending of the viewing angle of the X-ray detector, material thickness changes in this direction can be retrieved. As long as only one sample surface is altered, this gives reasonable data of the surface modification. However, X-ray imaging does not provide a clear reconstruction of the surface topology from the time during the melting process when both the front and back surfaces of the sample are modified.

For 3D surface detection, structured light imaging is a common option [13]. In the context of high-power laser processing, however, this is not applicable. The laser light can be removed by suitable filters. However, there can also be strong broadband light emissions, e.g., thermal emissions in the visible light range or from ignited plasma, which cannot be removed by filtering. This makes it difficult to evaluate the images correctly, as the structured illumination is easily overexposed by the process radiation. Interferometric measurements such as optical coherence tomography (OCT) overcome the problem of the process light emissions and are used, e.g., in a laser welding setup [14]. OCT has the advantage that the light emitted by the laser processing does not coherently interfere with the measurement illumination. However, 3D imaging requires scanning over the sample, which reduces the temporal resolution [15], especially for large sample areas, as required in this work.

Two-dimensional triangulation is an optical measurement technique which allows non-contact profile detection [16]. A line laser is directed onto the sample surface and the diffuse reflections are detected. Via optics, the reflection is projected onto a 2D pixel detector. A plane surface perpendicular to the triangulation scanner results in a straight line along one direction of the detector. Surface height alterations lead to a shifted position of the reflection position. Thereby, the surface profile along the complete laser line can be detected simultaneously. As modern sensor chips allow high frame rates, the surface profiles can also be detected with high temporal accuracy. This is particularly important for fast processes of melt dynamics. Red lasers are usually used in triangulation setups. However, this is not applicable for processes with samples of high temperature due to the thermal emission of the molten material at this wavelength. Therefore, it is necessary to move to wavelengths where there is no significant emission, as demonstrated with a laser with a wavelength of 405 nm for crack detection in metal production [17].

A comprehensive analysis of the laser perforation process requires a 3D observation of the surface topology. In order to move from a 2D triangulation setup to a 3D surface mapping, an additional lateral scanning of the measurement laser line is required. Different types of scanning systems have been developed for this purpose. One option is realized by linear motion of the sample or the triangulation system [18,19]. However, this type of setup is not well suited for the observation of the perforation process investigated here. Instead, lateral scanning via laser deflection can be performed [19,20,21]. The two-dimensional scanning of a point triangulation setup would be possible and result in a 3D surface map, but lacks an appropriate frame rate [21]. A sophisticated custom triangulation scanner is needed to scan only the measurement laser line via a galvanometer [20]. The simpler realization is to deflect both the laser line projected onto the sample and the reflected light going into the sensor [19].

The aim of this paper is to show that 2D triangulation is a suitable tool to monitor the sample surface deformation under high-power laser irradiation with large laser spots and to use the received data to describe the investigated process in detail. First, a bidirectional (front and back side) surface investigation is developed. With this, an evaluation of the change in surface profile and melt pool dynamics within the intersection slice is enabled. Second, with a further development of the scanning 3D triangulation setup shown by Schlarp et al. [19], a mirror symmetric deformation of the sample surface is shown. The improvement of this setup is that the tilting mirror is split into two parts, one for the deflection of the emitted measurement laser, which is located in the axis of rotation, and the second mirror is for the reflected light. This is necessary to allow a simultaneous perpendicular incidence of the high-power laser onto the sample and a near-perpendicular observation of the sample surface. The combination of both setups allow for a detailed observation of the different stages of the laser perforation process. Besides the pure existence, the times of occurrence are also detected. The retrieved information about surface deformation allows for a better understanding of the melt pool dynamics during high-power laser irradiation. The measured surface topology can also be used to evaluate laser reflection and scattering distributions in context of laser safety evaluations [22]. In addition, the measured data can be used as a benchmark for computer simulations calculating the effect of lasers on samples [23] and the melt pool dynamics during the laser perforation process [9].

## 2. Experimental Setup and Data Retrieval

To investigate the temporal change of the sample surface during high-power laser irradiation, 2D triangulation scanners were used in different settings. In Figure 1a, the general setup is shown with the bidirectional triangulation setup, while in Figure 1b, the specific setup for the scanning 3D triangulation setup is shown. The vertically oriented aluminum samples were irradiated almost perpendicular by a continuous-wave ytterbium fiber laser with a wavelength of 1070 nm, a line width of 5 nm, and a maximum average power of 10 kW (1% power stability). During the presented experiments, the laser power on the sample was 4 kW. The laser consists of 18 sub-modules, which are fed into the final multi-mode-working fiber with a core diameter of 200 μm, and exhibits a beam quality factor of $M^{2}=18$. The switching on/off time is below 40 μs. A zoom optics with a focal length between 5 m and 50 m allowed to change the spot size at the sample position. The laser spot shape was measured with a rotating measurement tip beam monitor at the position of the samples. The distance between the zoom optics and the samples was almost 5 m. The laser spot diameters (D4σ) used for the experiments were 16 mm, 22 mm, and 31 mm, respectively, with an almost circular Gaussian shape.

The aluminum samples (3.3547—EN AW 5083—Al Mg4.5Mn0.7 [24]) used had a thickness of 10 mm and a size of 200 mm × 200 mm. The surface finish corresponded to the technical quality, as provided by the supplier. The surface normal of the samples was horizontal and almost coaxial to the high-power laser.

Surface profiles of the samples were measured with 2D triangulation scanners (scanCONTROL LLT2900-100/BL, MICRO-EPSILON Messtechnik GmbH, Germany). They emit a laser line (wavelength 405 nm) onto the sample and determine the distance from the diffuse scattered light imaged by a 2D detector. To minimize disturbance of light emitted from the laser processing, bandpass filters were mounted in front of the scanner detector entries. The distance resolution is 12 μm and the lateral accuracy is between 64 μm and 97 μm depending on the sample-detector distance. The maximum frame rate is 300 Hz. The recording of the 2D triangulation scanners was started 0.5 s before the high-power laser to gain profiles of the unaltered sample surface. These profiles were used to remove a tilt between the sample and the scanner orientation. Additionally, the arbitrary values of the distance between the scanner and the sample was set to zero. Therefore, the presented surface profile values stated in this work represent the surface elevation. Hence, positive values represent an elevated surface while negative values represent a lowered surface with respect to the initial state. Some pixels with obvious measurement artifacts were removed during post-processing by a threshold algorithm. Such misinterpretations can be caused, for example, by steep profile edges, glare, or flying particles in the measuring laser path.

The back side of the sample was imaged with an optical camera with a frame rate of 50 Hz with single-frame triggering by a pulse generator. All devices were time-controlled by a delay generator. This allowed to start the triangulation scanners in advance to the laser. The perforation time of the samples was additionally detected by a photo diode, which captured the diffuse reflection of the laser beam on the beam dump after perforation. The signals of the laser control output and the photo diode were recorded with a 14-bit transient recorder with 1 kHz frame rate. A second beam dump was located at the angle of specular reflection due to the high amount of reflected laser power.

### 2.1. Bidirectional 2D Triangulation Setup

For the bidirectional 2D triangulation setup, two scanners were located at the front and back side of the sample, as shown in Figure 1a. The emitted measurement laser lines were oriented vertically. They were oriented in opposite direction to each other to achieve a common investigation plane. This plane intersected the sample at the center of the high-power laser spot. To allow an almost perpendicular laser processing, the 2D triangulation scanners were orientated off-axis to the high-power laser by an angle of around 10${}^{\circ }$. To prevent a misleading interaction of the two measurement lasers, they were operated in asynchronous mode. They were operated with a frame rate of 150 Hz.

### 2.2. Scanning 3D Triangulation Setup

A scanning setup with a deflection mirror was used for 3D imaging of the front of the sample. This setup represents an improved version of the one presented by Schlarp et al. [19], now allowing in situ measurements with the additional option of perpendicular laser processing. Based on optical considerations, a deflection of the measurement laser beam via a mirror is not changing the measurement principle of triangulation. It has only to be ensured that the emitted laser beam as well as the diffuse reflection of the sample get reflected by the mirror onto the sample and into the detector aperture, respectively. For a geometry with 90${}^{\circ }$ deflection, as indicated in Figure 1b, this requirement leads to the fact that only two small mirror areas are needed to perform the measurements. These two mirrors (front-side coated with aluminum) were mounted on a common U-shaped mounting support. This allowed the two mirrors to rotate around a common axis. In addition, the gap between the two mirrors enabled perpendicular sample processing with the high-power laser, with simultaneous coaxial triangulation investigation. The emitted measurement laser impinged on the first mirror near the edge to the hole. The rotation axis of the common mirror plane was aligned with this location of the emitted measurement laser on the first mirror. So, when the mirror is tilted, there is no change in the first laser deflection. The second mirror was placed at a position on the support where the diffuse sample reflection could be reflected into the scanner’s detector, while at the same time maximizing the distance between the two mirrors.

The distance between the mirror and the sample ($z_{ms}$) was approximately 270 mm, which corresponds to the far end of the detection range of the triangulation scanner. With a maximum tilt of the mirror of 4.5${}^{\circ }$, a range of 45 mm in the mirror tilt direction (horizontal) on the sample could be measured. In the vertical direction, the measuring range of the scanner was much larger than required at 90 mm. While the mirror was swept continuously back and forth, single exposures of the profile were recorded and the angular position of the goniometer used for tilting the mirror was recorded. Hence, the triangulation scanner could not be operated independently with a high repetition rate, such as for the bidirectional setup. The profile repetition rate was dependent on the cycling time of the control system and resulted in a profile repetition rate of around 43 Hz with an average 46 frames per tilt scan. Hence, an image frame rate of roughly 1 Hz was achieved.

The resulting data were: (i) the lateral position within the profile, (ii) the time stamp, (iii) the measured scanner–surface distance $z_{meas.}$ and (iv) the mirror tilt angle α. The measured scanner–surface distance $z_{meas.}$ was corrected to $z_{corr.}$ for the optical path length of the measurement laser corresponding to the specific measurement position by
(1)$$z_{corr.}=z_{lm}+\left(z_{meas.}-z_{lm}\right)\;\cdot \;\cos\left(2\alpha \right)\;\;,$$
with $z_{lm}$ being the distance between the triangulation scanner and the tilting mirror and α the tilt angle out of 45${}^{\circ }$ of the tilting mirror. From the mirror tilt angle α, on the other hand, the horizontal coordinates *y* were calculated by
(2)$$y=\left(z_{meas.}-z_{lm}\right)\;\cdot \;\tan\left(2\alpha \right)$$
in order to finally obtain 3D surface data. For both calculations, $z_{sm}$ and α were setup-dependent input variables. Subsequently, all tilts of the sample in horizontal and vertical direction were corrected on the basis of a primary scan before laser processing.

## 3. Results and Discussion

In laser penetration, the samples get heated by a high-power laser beam with spot diameters in the range of some centimeters. The location of interaction is heated until the molten material flows out, forming an almost cylindrical hole. To obtain a comprehensive picture of the penetration process, in situ surface profile measurements were performed using 2D triangulation scanners. To begin with, the surface profile dynamics of an exemplary laser penetration experiment are presented in detail. The measurements with changed parameters showed similar behavior with a different scaling in time. In a second step, an overview of the scaling for the other parameters is given without showing the surface profile changes again.

In Figure 2, the temporal evolution of the surface profiles of the exemplary high-power laser penetration process with a beam diameter of D4σ=22mm and a power of 4 kW is shown. The images show color-coded the surface elevation changes as a function of time, while positive values represent a protrusion and negative values represent indentations. In Figure 2a–c, the surface profile changes at the front side of the samples are shown in total, in addition to a close-up of the time when a first significant surface change starts, and a close-up of the time of perforation, respectively. The same time steps are shown in Figure 2e–g for the back side. In total, five different stages can be distinguished with a total of seven characteristic observations on both sides of the sample, whereby the back side is partly delayed in contrast to the front side. In the following, detailed explanations of the surface evolution are given point by point.

The sample surface does not change immediately after the laser is started. The sample surface has got a certain pristine roughness with a standard deviation of below 100 μm. It takes 13 s at the front side and 36.2 s at the back side until a first roughening of the surface is observed. This onset can be seen in Figure 2b,f by a change of the otherwise temporally constant surface structure (indicated by the yellow lines). This roughened area first starts in the center of the laser spot and enlarges with time, while the outer parts stay unchanged. This roughening is interpreted as only a texture modification and not a melting. Melting can hardly have taken place as also later such roughened areas are observed in clearly not-molten areas. For further discussion, see later;Shortly after the beginning of the surface roughening, at 16 s at the front and 42 s at the back side (orange lines in Figure 2), a strong surface profile change sets in. Again, beginning in the center of the laser spot, the surface starts to deform with an indentation in the top parts and a protrusion in the bottom parts. On the front side of the sample, this is more or less equally separated, while on the back side the indentation dominates and only a small protrusion is present. With time, the area of indentation and protrusion enlarges. The speed of change is thereby much faster at the back side than on the front side. This can be interpreted as the beginning of the surface melting at the front side (16 s and the melt through time (42 s)). The delayed melting on the back side can clearly be addressed to the finite thermal conductivity. The laser energy is deposited at the front side of the sample and must flow through the sample to the back side before a melting there can happen. Hence, the melting front must travel through the sample between 16 s and 42 s of laser irradiation. The late dominating indentation at the back side can be interpreted as a material flow towards the front;From time to time, very sudden changes in the size of the molten area can be identified, mainly on the front side but also on the back side(see Figure 2a,e, respectively). In the optical imaging (see Appendix A), one can see an orange glowing area, which increases with time. Hence, the area that has become so hot that it emits thermal radiation in the optical range becomes larger over time. In contrast to the naive assumption of a temporally constant temperature distribution, one can see a continuous and disordered movement of the hotspot within the melt pool. There is a clear temporal correlation between the increase in the size of the molten area determined by triangulation and the time at which the hot, glowing area at the border of the molten pool is near the position of the triangulation profile. Even in videography, the sudden enlargement of the melt pool in such places can sometimes be seen. Consequently, the stepwise enlargement of the melt pool can be interpreted as preferential melting at this location due to its close vicinity to a very hot molten fraction;After 63.1 s of laser irradiation, the surface profiles start to change much faster than before (Figure 2c,g red lines). At the front side, the protrusion enlarges strongly (from 7 mm to 14 mm height) and starts to flow downwards fast, while the indentation at the top also grows (from 3 mm to 5.5 mm depth). Simultaneously, on the back side only the indentation enlarges (from 3.3 mm to 5.4 mm depth), while the protrusion decreases slightly (from 1.2 mm to 1.0 mm height). This can be interpreted as the melt flowing from the back to the front. It marks the beginning of the final flow-out phase;At 63.46 s after the start of laser irradiation (magenta lines in Figure 2), the triangulation scanners start to observe large areas of significantly wrong distance values or even no detected values at all. This coincides with the time when the photodiode aligned at the beam dump behind the sample registers a significant increase in scattered light. Hence, this is the time of perforation. The resulting hole enlarges fast and stabilizes in size within 0.2 s;Until 64.42 s, the laser was turned on (black lines in Figure 2). Shortly after perforation, the triangulation scanners still register certain profile data within the hole area. However, most of them have unrealistic values (much too far or too close). This can be attributed to the fact that either significant amounts of vapor or flames appear immediately after perforation (see also Appendix A). Furthermore, small aluminum droplets are in the hole area, which are heated easily to such high temperatures that they even emit thermally at the triangulation laser wavelength of 405 nm, resulting in inaccurately interpretable data for the scanners. These strongly misleading values reduce with time and vanish shortly after the laser is turned off. Then, a stable hole with the size of around 18.5 mm establishes.

Effort has been made to also measure 3D triangulation data. With the bidirectional 2D triangulation setup, many aspects of the laser penetration process can be investigated. However, only one cross-section through the sample can be detected. The areas outside this cross-section are still missing. To also obtain information about these outer areas, the scanning 3D triangulation setup has been designed. In Figure 3, three surface profile maps of the front side of the sample at different times are shown. The results of the bidirectional 2D triangulation with a delayed melting, the development of indentation and protrusion, and fast hole formation are observed. The 3D information provides confidence that extrapolation to surrounding areas is possible from knowledge of the central vertical profile. An almost-round and mirror-symmetric melt pool region develops after a certain delay of unchanged sample surface. The indentation and protrusion are strongest in the central vertical line, both near the upper and lower border of the melt pool, respectively. In the horizontal direction, the surface elevation (indentation as well as protrusion) decreases symmetrically until it reaches the unchanged state outside the melt pool. The non-circular shape (prominent in Figure 3b) is again attributed to the observed stepwise enlargement of the melt pool in cases where hot molten material flows into the outer region of the melt pool.

It should be noted that scanning 3D triangulation is equivalent to rolling shutter videography. Hence, the resulting surface map does not represent a certain point in time. The scanning speed was around 1 Hz, resulting in a temporal blurring of around 1 s between the left and right side of the images. With the results of the bidirectional 2D triangulation of Figure 2 in mind, this repetition rate seems to be sufficient for imaging the melt pool dynamics. Only the short time period of the final perforation (accelerated flow downwards and hole formation) is too fast to be imaged properly with the scanning 3D triangulation.

The 2D triangulation data can be evaluated in time by integrating over each surface profile. Since the triangulation scanners only measure a 2D profile, the bidirectional 2D setup can only determine the material change within the examination slice that could be called the “change of material cross-section”. In Figure 4, the integrated value of each profile is plotted as a function of time for the front and back side as well as their sum. Positive values represent an elevation of the sample surface towards the detector and vice versa.

It is observed that there is a slight increase of the material cross-section already before the onset of melting (orange lines) on both sides. However, only on the front side the material cross-section increases until the perforation at 63.46 s (magenta line). On the back side however, after the onset of melting it starts to decrease again, being slightly below zero before perforation.

Although a detailed analysis of the exact processes that have an influence on the surface dynamics during laser irradiation is beyond the scope of this paper, a few aspects for the interpretation of the change of material cross-section will be mentioned in the following. The initial increase of sample thickness within the laser spot area (represented in Figure 4 as an increase in material cross-section) before the surface melts can be attributed to some extent to thermal expansion. When the laser deposits energy in the sample, this heats up. With that, the material expands. The coefficient of thermal expansion is $23.4\;\times \;10^{-6}$ K^−1^ for the EN AW-5083 aluminum alloy used [25]. With a sample thickness of 10 mm and a temperature difference of 600 K (roughly room temperature to liquidus temperature), this would result in a thickness increase of 140 μm. However, the observed thickness change, represented as surface elevation increase, is around two times larger than this value. Therefore, other processes must be considered in addition. However, a detailed investigation of this point is beyond the scope of this work.

The further increase of the material cross-section at the front side might to some extent originate from a liquid flow from the outer areas into the line of investigation. After the onset of surface melting at the back side, a decrease in material cross-section is observed, which clearly can be attributed to a flow of the liquid aluminum to the front side. This interpretation is supported by the fact that the global material cross-section almost saturates after the onset of surface melting at the back side (second orange line in Figure 4). Somewhat contradictory to this is the fact that the integrated material surface change of the scanning 3D triangulation measurements, i.e., the change of material volume, shows a similar constant increase for the front side until perforation (data not shown). There, a pure delocalization of material along the sample surface should have no influence. The increase in material cross-section between the melting of the front surface and the melting of the rear surface must therefore be due not only to the fluid flow but also, for example, to thermal expansion.

The observed stepwise enlargements of the molten sample area (see Figure 2) do not correlate with the characteristic dynamics of the material cross-section. This allows the interpretation that the stepwise enlargement is mainly a shift of material within the cross-section plane of the 2D triangulation measurement, not perpendicular to it. After the laser was turned off (64.42 s), a stable material cross-section establishes for both measurement sides with a comparable value of roughly 200 mm^2^. This fits well to the measured hole diameter of around 18.5 mm and a material thickness of 10 mm.

The ex situ measurement of the resulting hole shown in Figure 5 reveals that the hole is close to elliptical and the major axis is horizontal. It should be noted that the flowed-out melt pool solidified as a bubble below the hole due to a setup protection plate under the sample. Furthermore, a roughened ring around the hole is observed optically and also in the surface profile measurement. From Figure 2 and Appendix A, one can conclude that almost all molten material flows out during perforation. The final top edge of the hole is located where the indentation started. The final bottom edge of the hole is located roughly between the beginning of the protrusion at the front and back side. Therefore, it is assumed that the roughened area around the hole cannot have exceeded the liquidus temperature. This roughening process must be an effect taking place below the liquidus temperature, but maybe above the solidus temperature. Significant differences in backscattered electron contrast are observed in scanning electron microscopy (data not shown), possibly due to phase separation of the individual alloy constituents in this roughened region. However, a detailed analysis is beyond the scope of this paper.

Until now, one representative example was shown. In the following, the different aspects in the context of a varied spot diameter will be described. In Table 1, the perforation times for the three repetitions of the three different spot diameters used are shown. Despite a certain fluctuation within the experiments with the same parameters, a correlation of the perforation time with the spot diameter is observed. Without claiming to be exhaustive, two aspects are listed here to explain the fluctuations of the perforation time within one parameter set. First, to a small extent, different reflection rates at different sample positions could lead to different losses, at least at the beginning of laser penetration. secondly, the dynamics of the melt pool is affected by instabilities and partly unpredictable and chaotic behavior and thus cannot be expected to be perfectly repeatable in the experiments. This means that the different breakup of the melt pool film on the top side can lead to significant changes in the perforation time.

The resulting hole sizes were comparable for all experiments with a material cross-section change after perforation ranging from 200 mm^2^ to 250 mm^2^ (measured at the front side). Therefore, the amount of material flowed out can be considered to be comparable. Consequently, while the laser power remains constant and only the spot size is varied, the net heating rate of the material that flowed out must decrease with increasing perforation time (at least if a constant absorptivity is assumed). This must be attributed to a higher heat loss in the laser impact zone due to thermal conduction. Besides the perforation time, the times of occurrence of the other observed characteristic steps within the process of laser penetration also do correlate with the spot size.

It can be observed that the final hole is located slightly above the laser spot. This dislocation effect is stronger the smaller the laser spot and the higher the laser intensity (see Appendix A); while the laser hits both the indentation and the protrusion largely symmetrically at the larger spot diameters (see Figure 2), the smallest spot diameter of D4σ=16 mm mainly hits the liquid aluminum protrusion bubble in the last third of the indentation process. For the larger spot diameters, the position of the unchanged surface elevation between the protrusion and the indentation moves only slightly upward (see Appendix A). For the smallest spot diameter of D4σ=16 mm, however, this position shifts strongly upward by about 10 mm. It can also be observed that the indentation area has a comparatively smaller material thickness over a longer period of time for the smallest spot diameter.

Without claiming to be comprehensive, this different melting behavior can be interpreted in the following way. The indentation and protrusion areas of the melt pool are caused by gravity. The high surface tension of liquid aluminum [26] compared to other materials [27] leads to the formation of a comparably large protrusion before the perforation. Hence, for other materials, smaller protrusions and also smaller melt pools with subsequently smaller holes are expected. For the perforation of these samples (aluminum EN AW 5083 with 10 mm thickness), the results show that a hole diameter of around 20 mm seems to be required for the melt pool to flow out. Consequently, the two large spots are large enough to directly heat the entire area required while the smallest spot can only interact with a small part of the required melt pool, albeit with a higher intensity. The hot material experiences a convection-driven upwards flow against the gravitational force within the melt pool. Subsequently, the melt pool enlarges at the top. Furthermore, with the growing protrusion at the bottom, the heating of the material at the bottom-back side gets more and more inefficient as it only can occur by thermal conduction. This strong separation of hot, upward flowing material within the molten pool, and the resulting, somewhat protected, bottom of the interaction zone from melting, is less pronounced for large point sizes.

Further aspects for a changed melting dynamics might be multiple reflections. They can lead to a delocalized energy input, even at positions outside the primary laser spot. This can take place preferentially at strongly curved surfaces, which especially form shortly before the perforation.

## 4. Conclusions

In this paper, the applicability of the 2D triangulation for in situ surface profile measurements in high-power laser penetration processes has been demonstrated. With the developed bidirectional 2D triangulation setup, the change of the surface profile at the front and back side of the sample was measured. This allowed to identify several specific steps in the laser penetration process, such as surface melting, melt pool development and dynamics, and sample perforation. Such clear observations of the material surface dynamics have not been possible with other techniques until now. Not only the occurrence of the different steps in the laser penetration process could be observed. The times of their occurrence could also be determined. Despite some differences for varying laser spot sizes, the penetration process is mainly the same, with only a temporal scaling. With an additional scanning 3D triangulation setup with a tilting mirror a mirror-symmetric melt pool formation has been shown. It was also possible to show that extrapolation from a measurement of the central profile to the outer regions is permissible. With this demonstration of the applicability of 2D triangulation to the laser penetration process, the way for a wide variety of further investigations has been paved. Further analysis of the phenomena in laser penetration depending on process parameters such as spot size, laser power, and air flow are obvious.

## Figures and Tables

**Figure 1 materials-15-03743-f001:**
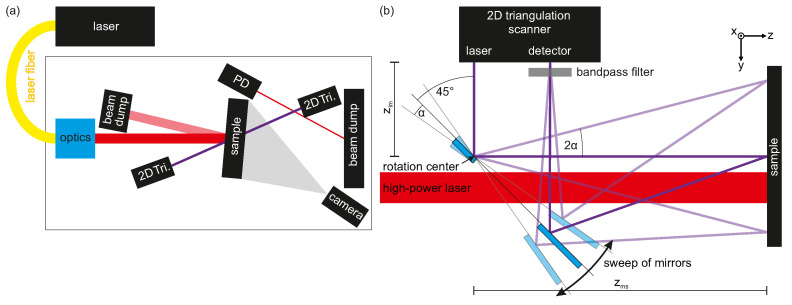
Illustration of the experimental setups: (**a**) the complete setup with the bidirectional triangulation measurement and (**b**) specific scanning 3D triangulation setup. In the latter, the measurement laser light was deflected by mirrors (light blue), which were rotated around a common axis, allowing to scan the sample surface perpendicular to the 2D triangulation profile.

**Figure 2 materials-15-03743-f002:**
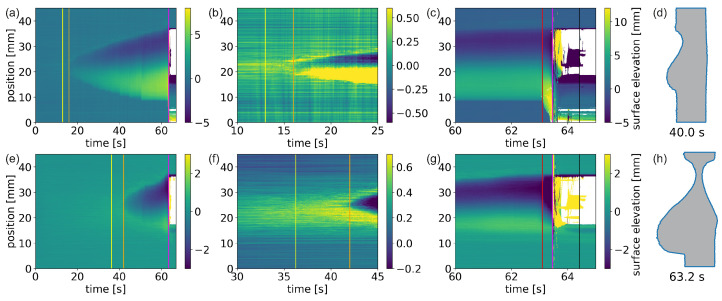
Temporal surface change at the front side (**a**–**c**) and back side (**e**–**g**) of an aluminum plate irradiated with a laser of 4 kW power and a spot diameter of D4σ=22mm. The vertical lines represent: yellow—start of surface roughening; orange—start of melting; red—start of accelerated material downward flow; magenta—perforation; black—laser turn off. In (**d**,**h**), cross-sections at two selected times are shown.

**Figure 3 materials-15-03743-f003:**
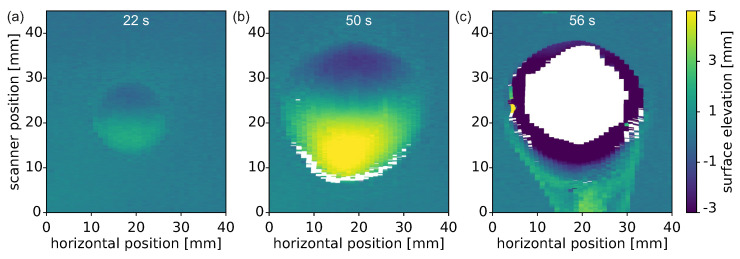
3D surface plots of the front side of an aluminum plate of three selected times while irradiated with a laser of 4 kW power and a spot diameter of D4σ=22mm. The evolution of a growing indentation at the top and a growing protrusion are clearly visualized (**a**,**b**) before perforation (**c**).

**Figure 4 materials-15-03743-f004:**
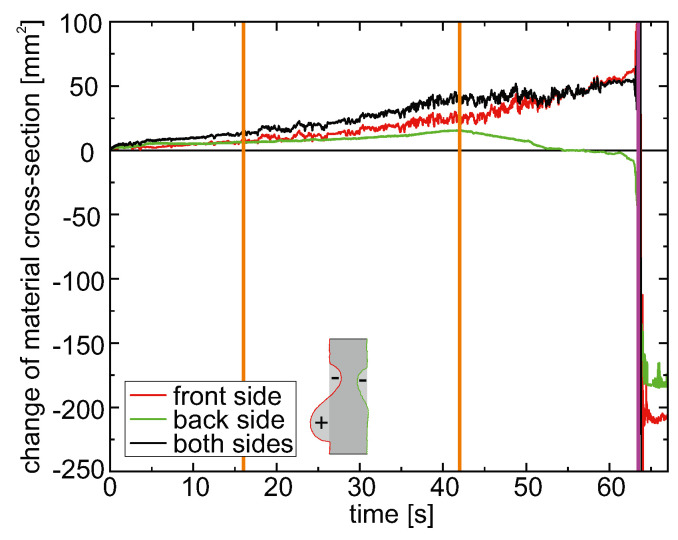
Temporal change of the material cross-section for the experiments in Figure 2. For both sample sides at the beginning an increasing material thickness is observed. For the front side this holds true until the perforation (magenta line), while for the back side, the sample thickness increases only until the melting (orange line, 42 s), after which it decreases again. The global material cross-section increases until perforation.

**Figure 5 materials-15-03743-f005:**
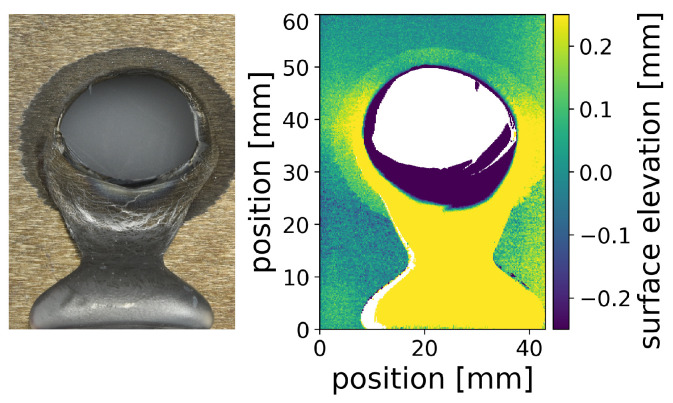
Images of the front side of the sample of Figure 2 after laser perforation, as photograph and as 3D surface profile. Besides the hole in the center and the flow-out material bubble at the bottom, the heat-affected area around the hole is visible.

**Table 1 materials-15-03743-t001:** Perforation times for three repetitions of laser penetration with three different spot diameters D4σ. The intensity is the calculated mean value within the 86% criteria.

D4σ	Intensity	Perforation Time			Mean Perforation Time
[mm]	[$W/{\mathrm{mm}}^{2}$]	[s]			[s]
16	17.1	47.2	51.8	53.2	50.7 2.6
22	9.0	63.5	85.1	84.0	77.5 9.9
31	4.6	77.2	95.9	94.1	89.1 8.4

## Data Availability

The data presented in this study are available on request from the corresponding author.

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
