# Peer review of "2D and 3D Triangulation Are Suitable In Situ Measurement Tools for High-Power Large Spot Laser Penetration Processes to Visualize Depressions and Protrusions before Perforating"

_materials, 2022, doi:10.3390/ma15113743_

Round 1

Reviewer 1 Report

Authors have used precise approaches like surface melting, indentations, and protrusions during melt pool development and their dynamics, and the perforation was visualized. The measurements showed a mirror-symmetric melt pool and the possibility to extrapolate from the central profile to the outer regions in most situations. The article is interesting. The following points need to be answered carefully for the improvement of the manuscript.

  1. Remove the word ‘we’ throughout the thesis
  2. Mention clearly novelty of the research.
  3. Provide complete specification of EN AW 5083 - Al Mg4,5 Mn0,7 aluminium sample
  4. Authors mentioned that “This roughening is interpreted as only a texture modification”. How it is valid?
  5. How 3D triangulation data has been measured?
  6. Need more explanation on fig. 2
  7. The authors stated ‘We attribute the initial increase in material cross-section before the surface melts to some extent to thermal expansion. Clarify the sentence.
  8. Conclusions should be rewritten. Remove citations from the conclusion section.

Author Response

Dear Reviewer,

we want to thank you for your time to review out manuscript. And thanks for your valuable comments with which we now could improve the manuscript. Please find below our answers to your comments. The changes made in the manuscript are explained here, the changes you can find in the change-tracked new submission.

Authors have used precise approaches like surface melting, indentations, and protrusions during melt pool development and their dynamics, and the perforation was visualized. The measurements showed a mirror-symmetric melt pool and the possibility to extrapolate from the central profile to the outer regions in most situations. The article is interesting. The following points need to be answered carefully for the improvement of the manuscript.

  1. Remove the word ‘we’ throughout the thesis
    We see this as a personal style issue. It is quite common in literature to publish with ‚we‘. However, to follow your comment we have rewritten all sentences in the text containing ‚we‘. See the multiple changes in the change tracked manuscript.

  2. Mention clearly novelty of the research.
    We are sorry, that we did not clear enough mention the novelty in the paper. We have add points to the abstract and also rewritten a part of the Introduction. We hope that we now have pointed out the novelty to your wishes. See multiple changes in the change tracked manuscript.

  3. Provide complete specification of EN AW 5083 - Al Mg4,5 Mn0,7 aluminium sample
    We have improved the naming of the sample specification to 3.3547 - EN AW-5083 – Al Mg4.5 Mn0.7 and also have cited the DIN EN norm.

  4. Authors mentioned that “This roughening is interpreted as only a texture modification”. How it is valid?
    We know, that at this position in the paper, no comprehensive discussion about this topic is given. However, we have already mentioned in the manuscript, that this topic is discussed later in the manuscript in more details. But for a bit more clarity at this beginning of the results section, we have added a comment, why we can state, that this area is not yet molten:
    This roughening is interpreted as only a texture modification and not a meltingas discussed later.. Melting can hardly have taken place as also later such roughened areas are observed in clearly not molten areas. For further discussion, see later.

  5. How 3D triangulation data has been measured?
    In section 2.2, the scanning 3D triangulation setup was described. First the building of the setup with the divided tilting mirror, the common rotation axis of these two mirros and the reason why this setups is possible is described. Then, also the rough dimensions of the setup are mentioned. And finally also the data processing to gain positional data out from angles were shown. Therefore, we don’t know from this short comment, which details you are missing. We have carefully read this section and have added points which hopefully help the reader to perfectly understand the measurement scheme. Also, we have improved Fig. 1 (b) for more clarity.

  6. Need more explanation on fig. 2
    We are a somewhat wounded that Fig. 2 is not explained well accurately in your eyes. There are 1.5 pages of detailed explanation about the many points, which can be observed in Fig. 2. Furthermore, some details are even more explained later in the paper with clearly referencing Fig. 2. Unfortunately, we can't tell from this simple comment exactly which points you think were not explained well enough.
    However, we tried to add some more informations in the manuscript text. First, we have added two representative cross-sectional images in Fig. 2 for a more intuitive understanding of the content of Fig. 2 . They were until now only shown in the grafical abstract.
    See the multiple changes done in the revised version. We hope with that we could fulfill your needs.

  7. The authors stated ‘We attribute the initial increase in material cross-section before the surface melts to some extent to thermal expansion. Clarify the sentence.
    As shown in Fig. 4, the material cross-section changes already before the onset of real surface melting occurrs. This point we wanted to adress with this sentence. We are sorry, that we could not point clearly, what our, at the moment, limited interpretation is. We have added more information in this paragraph which hopefully clarify is:
    We attribute tThe initial increase in material cross-section before the surface melts can be attributed to some extent to thermal expansion. When the laser deposits energy in the sample this heats up. With that the material expands. The coefficient of thermal expansion is 23.4 × 10−6 K−1 for the EN AW-5083 aluminium alloy used [25]. With a sample thickness of 10 mm and a temperature difference of 600 K (roughly room temperature to liquidus temperature) this would result in a thickness increase of 140 µm. However, the observed thickness change, represented as surface elevation increase, is around two times larger than this value, so that. Therefore, other processes must be considered in addition. However, a detailed investigation of this point is beyond the scope of this work.

  8. Conclusions should be rewritten. Remove citations from the conclusion section.
    We agree with you that it was a mistake on our part to cite papers in the conclusion. We have removed them.
    On the other hand, we do not completly know, why you think we should rewrite the conclusion? We have added some points concerning the clarification of the novelty of the paper and have removed the ‚we‘. We hope that this were the points you were adressing. If not, your very short comment was maybe not precise enough.

Reviewer 2 Report

The article presents interesting issues concerning applicability of 2D and 3D triangulation for surface topology observations.

The described analysis of the results of the conducted research will also certainly be helpful for scientists in the field of materials engineering and laser processing. However, the article requires some changes prior to publication, such as:

  1. The Introduction in Lines 71-75 should be the last Introduction paragraph.
  2. The Introduction paragraph should also contain information about the novelty in the article and the purpose of the research (i.e. what the research work was trying to achieve or check).
  3. The descriptions under the Figures are too long, please put short and concise descriptions below the Figures, while a longer description for a given Figure should be placed in the text, where there is a reference to this Figure and its analysis.
  4. Figures separate a continuous text in several places in the text. For the correct layout of the article, Figures should not separate continuous text.

Author Response

Dear Reviewer,

we want to thank you for your time to review out manuscript. And thanks for your valuable comments with which we now could improve the manuscript. Please find below our answers to your comments. The changes made in the manuscript are explained here, the changes you can find in the change-tracked new submission.

The article presents interesting issues concerning applicability of 2D and 3D triangulation for surface topology observations.

The described analysis of the results of the conducted research will also certainly be helpful for scientists in the field of materials engineering and laser processing. However, the article requires some changes prior to publication, such as:

  1. The Introduction in Lines 71-75 should be the last Introduction paragraph.
    We like your suggestion of changing this part of the Introduction. We have rewritten this part completly. See multiple changes in the change tracked manuscript.

  2. The Introduction paragraph should also contain information about the novelty in the article and the purpose of the research (i.e. what the research work was trying to achieve or check).
    We are sorry, that we were not able to clearly point out the aims and novelties of this article. We have therefore added points to the abstract as well as the Introduction. We hope, that we now have adressed all required points you were missing. See multiple changes in the change tracked manuscript.

  3. The descriptions under the Figures are too long, please put short and concise descriptions below the Figures, while a longer description for a given Figure should be placed in the text, where there is a reference to this Figure and its analysis.
    We agree with you that captions should not be unnecessarily long. But we think that at least a rough description of all main points of a figure should be given. We have tried to shorten the captions as much as possible because we feel that a reader does not necessarily need to read the text to have a basic understanding of the illustrations.

  4. Figures separate a continuous text in several places in the text. For the correct layout of the article, Figures should not separate continuous text.
    You are absolutely right that we had unfortunately done this wrong. The MDPI template says that the fugures as well as tables should always be placed directly in their place. We have of course corrected this.

Round 2

Reviewer 1 Report

The authors have answered the comments systematically. The article is very interesting. Now the paper is accepted and recommended for publication.